# New Radical-Cation Salts Based on the TMTTF and TMTSF Donors with Iron and Chromium Bis(Dicarbollide) Complexes: Synthesis, Structure, Properties [†]



Denis M. Chudak [1], Olga N. Kazheva [2,3,*], Irina D. Kosenko [4], Gennady V. Shilov [2], Igor B. Sivaev [4,5], Georgy G. Abashev [6], Elena V. Shklyaeva [6], Lev I. Buravov [2], Dmitry N. Pevtsov [2], Tatiana N. Starodub [7], Vladimir I. Bregadze [4] and Oleg A. Dyachenko [2]

[1] Chemistry Department, V. N. Karazin Kharkiv National University, 4 Svoboda Sq., 61077 Kharkiv, Ukraine; chudakdenis@gmail.com
[2] Institute of Problems of Chemical Physics, Russian Academy of Sciences, 1 Semenov Av., 142432 Chernogolovka, Moscow Region, Russia; genshil@icp.ac.ru (G.V.S.); buravov@icp.ac.ru (L.I.B.); pevtsovdm@gmail.com (D.N.P.); doa@rfbr.ru (O.A.D.)
[3] Institute of Experimental Mineralogy, Russian Academy of Sciences, 4 Academician Osypyan Str., 4, 142432 Chernogolovka, Moscow Region, Russia
[4] A.N. Nesmeyanov Institute of Organoelement Compounds, Russian Academy of Sciences, 28 Vavilov Str., 119991 Moscow, Russia; kosenko@ineos.ac.ru (I.D.K.); sivaev@ineos.ac.ru (I.B.S.); bre@ineos.ac.ru (V.I.B.)
[5] Basic Department of Chemistry of Innovative Materials and Technologies, G.V. Plekhanov Russian University of Economics, 36 Stremyannyi Line, 117997 Moscow, Russia
[6] Organic Chemistry Department, Perm State University, 15 Bukirev Str., 614990 Perm, Russia; gabashev@psu.ru (G.G.A.); EV_Shklyaeva@psu.ru (E.V.S.)
[7] Institute of Chemistry, Jan Kochanowski University, 15G Swietokrzyska Str., 25-406 Kielce, Poland; tstarodub@ujk.edu.pl
* Correspondence: koh@icp.ac.ru
[†] Dedicated to Professor Alan J. Welch in recognition of his outstanding contribution to the chemistry of carboranes.

**Abstract:** New radical-cation salts based on tetramethyltetrathiafulvalene (TMTTF) and tetramethyltetraselenefulvalene (TMsTSF) with metallacarborane anions (TMTTF)[3,3′-Cr(1,2-$C_2B_9H_{11}$)$_2$], (TMTTF)[3,3′-Fe(1,2-$C_2B_9H_{11}$)$_2$], and (TMTSF)$_2$[3,3′-Cr(1,2-$C_2B_9H_{11}$)$_2$] were synthesized by electrocrystallization. Their crystal structures were determined by single crystal X-ray diffraction, and their electrophysical properties in a wide temperature range were studied. The first two salts are dielectrics, while the third one is a narrow-gap semiconductor: $\sigma_{RT} = 5 \times 10^{-3}$ Ohm$^{-1}$cm$^{-1}$; $E_a \approx 0.04$ eV (aprox. 320 cm$^{-1}$).

**Keywords:** iron bis(1,2-dicarbollide); chromium bis(1,2-dicarbollide); tetramethyltetrathiafulvalene; tetramethyltetraselenafulvalene; radical-cation salts; crystal and molecular structure; electric conductivity

## 1. Introduction

Radical-cation salts and charge transfer complexes based on derivatives of tetrathiafulvalene (TTF) constitute a wide class of organic materials with transport properties ranging from insulating to superconducting [1–4]. This work is part of the systematic study of radical-cation salts of tetrathiafulvalene and its derivatives with metallacarborane anions, of which earlier results were summarized in works [5–7].

Transition metal bis(dicarbollide) complexes [3,3′-M(1,2-$C_2B_9H_{11}$)$_2$]$^-$ (M = Fe, Co, or Ni) are of great interest as counterions for the synthesis of TTF-based molecular conductors due to the unique high stability, possibility of tuning the charge and nature of the metal, and wide range of options for modification with dicarbollide ligands via hydrogen substitution by other atoms and functional groups [5,6]. Although most of the compounds studied were BEDT-TTF-based radical-cation salts, recently, we have synthesized radical-cation

salts based on such unconventional and rather exotic donors as bis(1,3-propylenedithio)-tetrathiafulvalene [8,9], dibenzotetrathiafulvalene [10], and 4,5-ethylenedithio-4′,5′-(2-oxa-1,3-propylenedithio)-tetrathiafulvalene [9]. On the other hand, although compounds of the composition $(TMTXF)_2Y$ (X = T, S) are usually classical organic metals among which the first organic superconductors were discovered [4,7], and TMTTF and TMTSF radical-cation salts continue to attract the attention of researchers [11–15], very little attention has been paid to TMTTF and TMTSF radical-cation salts with metallacarborane anions [16–19]. This prompted us to prepare and investigate new TMTTF and TMTSF radical-cation salts with metallacarborane anions.

This contribution describes the synthesis, structure, and electrical conductivity of new salts with TMTTF and TMTSF radical-cations and metallacarborane anions: $(TMTTF)[3,3′-Cr(1,2-C_2B_9H_{11})_2]$ (**1**), $(TMTTF)[3,3′-Fe(1,2-C_2B_9H_{11})_2]$ (**2**), and $(TMTSF)_2[3,3′-Cr(1,2-C_2B_9H_{11})_2]$ (**3**).

## 2. Results and Discussion

Single crystals of compounds **1**–**3** suitable for X-ray diffraction studies in the form of thin plates were obtained by electrochemical crystallization (See Supplementary Materials and Table 1). The crystal structure of **1** is formed by the TMTTF radical-cations and the $[3,3′-Cr(1,2-C_2B_9H_{11})_2]^-$ anions occupying general positions in the unit cell (Figure 1). $(TMTTF)[3,3′-Cr(1,2-C_2B_9H_{11})_2]$ has a pseudo-layered structure, in which anionic layers alternate along the *ac* diagonal with layers formed by radical-cation dimers (Figure 2). The dimer formation corresponds to the stoichiometry of the salt: in this case due to the Peierls instability a phase transition should occur with doubling of the stacks period [7]. The distances between the averaged planes of the TMTTF donors in the dimers are 3.38 Å (the planes are drawn through all S atoms), and the dihedral angle between the planes is 0° by symmetry conditions. There are short intermolecular S . . . S interactions (3.426(1)–3.432(1) Å) of the "face-to-face" type between the TMTTF donors in the dimers.

**Table 1.** Crystal data and structure refinement for $(TMTTF)[3,3′-Cr(1,2-C_2B_9H_{11})_2]$ (**1**), $(TMTTF)[3,3′-Fe(1,2-C_2B_9H_{11})_2]$ (**2**), and $(TMTSF)_2[3,3′-Cr(1,2-C_2B_9H_{11})_2]$ (**3**).

| Compound | (1) | (2) | (3) |
|---|---|---|---|
| Empiric formula | $C_{14}H_{34}B_{18}CrS_4$ | $C_{14}H_{34}B_{18}FeS_4$ | $C_{24}H_{46}B_{18}CrSe_8$ |
| Formula weight | 577.23 | 581.08 | 1212.87 |
| Crystal system | Monoclinic | Monoclinic | Triclinic |
| Space group | $P2_1/c$ | $C2/m$ | $P\,1$ |
| $a$ (Å) | 11.726(2) | 17.3487(8) | 7.451(4) |
| $b$ (Å) | 12.753(2) | 12.0235(6) | 12.342(6) |
| $c$ (Å) | 19.387(3) | 6.6791(3) | 12.961(7) |
| $\alpha$ (°) | 90 | 90 | 117.743(7) |
| $\beta$ (°) | 102.701(2) | 90.7840(6) | 92.344(8) |
| $\gamma$ (°) | 90 | 90 | 100.325(8) |
| $V$ (Å$^3$) | 2828.3(6) | 1393.08(11) | 1027.1(9) |
| $Z$ | 4 | 2 | 1 |
| $\lambda$ (Å) | 0.71073 | 0.71073 | 0.71073 |
| $D_{calc}$ (Mg m$^{-3}$) | 1.36 | 1.38 | 1.96 |
| $\mu$ (mm$^{-1}$) | 0.708 | 0.850 | 7.388 |
| Number of reflections collected | 28470 | 11191 | 4513 |
| Number of independent reflections | 8147 | 2319 | 4513 |
| Number of reflections with $[F_0 > 4\sigma(F_0)]$ | 6787 | 2183 | 3754 |
| Number of parameters refined | 426 | 130 | 233 |
| $(2\theta)_{max}$ (°) | 60.48 | 63.70 | 55.44 |
| $R$ | 0.037 | 0.021 | 0.051 |

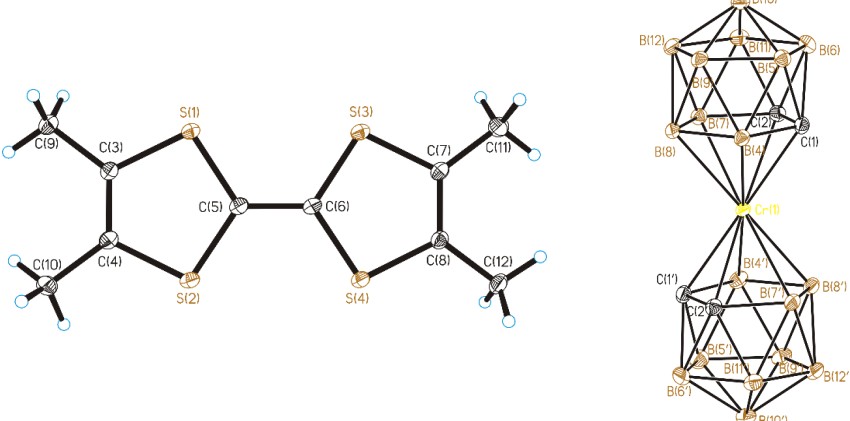

**Figure 1.** TMTTF radical-cation and anion in (**1**). Thermal ellipsoids are given at 30% probability level. Cage H atoms omitted for clarity.

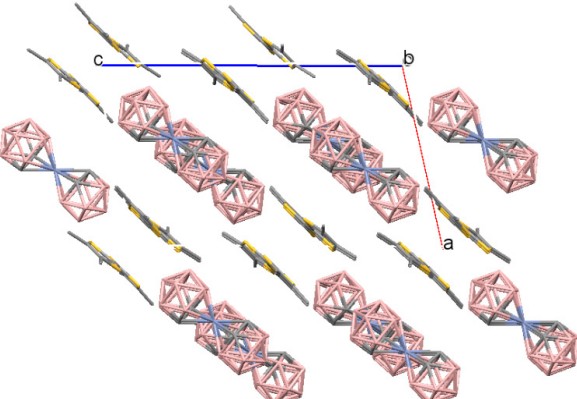

**Figure 2.** Crystal packing fragment of (**1**). A view along the *b* axis. The unit cell is outlined. H atoms are omitted for clarity.

The TMTTF$^+$ radical-cations are non-planar and have a "boat" conformation: the maximum deviations of terminal C(9), C(10), C(11), and C(12) atoms from the plane of the averaged molecule drawn through all sulfur atoms are 0.30–0.36 Å.

The Cr-C and Cr-B bond lengths are 2.173(2)–2.180(2) and 2.232(2)–2.279(2) Å, correspondingly. The distances from the chromium atom to the C$_2$B$_3$ faces of the dicarbollide ligands are equal to 1.68 Å, which is close to the corresponding distances found in the structures of Cs [3,3′-Cr(1,2-C$_2$B$_9$H$_{11}$)$_2$] [20], (TTF)[3,3′-Cr(1,2-C$_2$B$_9$H$_{11}$)$_2$] [21], and (BEDT-TTF)$_2$[3,3′-Cr(1,2-C$_2$B$_9$H$_{11}$)$_2$] [22,23]. The dicarbollide ligands in the [3,3′-Cr(1,2-C$_2$B$_9$H$_{11}$)$_2$]$^-$ anion are turned relative to each other by 180°, forming the *transoid* conformation. The C$_2$B$_3$ faces deviate slightly from parallel, being inclined by 178.7° to each other.

The electrical conductivity measurements have shown that 1 is an insulator with $\sigma_{293} \sim 10^{-11}$ Ohm$^{-1}$cm$^{-1}$. The low value of electrical conductivity is apparently connected with the absence of conducting layers and dimerization of the radical-cations stacks.

It should be noted that compound 1 is the first TMTTF radical-cation salt with an unsubstituted transition metal bis(dicarbollide), while the radical-cation salts (TMTTF)[8-HO-3,3′-Co(1,2-C$_2$B$_9$H$_{10}$)(1′,2′-C$_2$B$_9$H$_{11}$)] and (TMTTF)(8,8′-Cl$_2$-3,3′-Co(1,2-C$_2$B$_9$H$_{10}$)$_2$)$_2$ obtained earlier contained substituted bis(dicarbollide) anions [16,17].

The crystal structure of (TMTTF)[3,3′-Fe(1,2-C$_2$B$_9$H$_{11}$)$_2$] (**2**) is formed by a quarter of the TMTTF radical-cation in a special position placed on the *m* plane and a quarter of the [3,3′-Fe(1,2-C$_2$B$_9$H$_{11}$)$_2$]$^-$ anion in the 2/*m* special position of the unit cell (Figure 3). The compound **2** is characterized by a structure where the TMTTF cations and the metallacarborane anions form staggered stacks (Figures 4 and 5). The distances between the averaged

planes of the TMTTF donors in the dimers are 3.38 Å, and the dihedral angle between the planes is 0° by symmetry conditions.

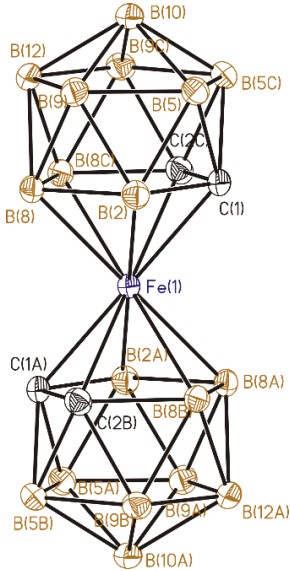

**Figure 3.** Anion in (**2**). Thermal ellipsoids are given at 30% probability level. Cage H atoms omitted for clarity.

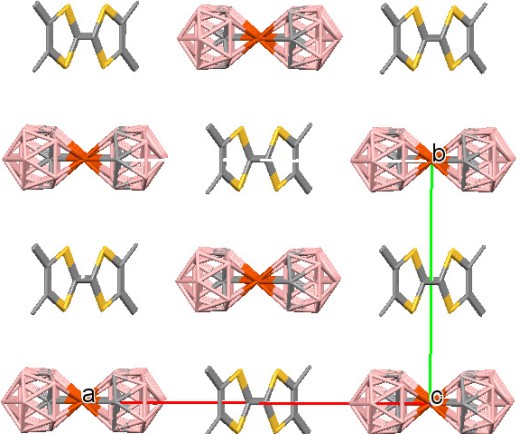

**Figure 4.** Crystal packing fragment of (**2**). A view along the *c* axis. The unit cell is outlined. H atoms are omitted for clarity.

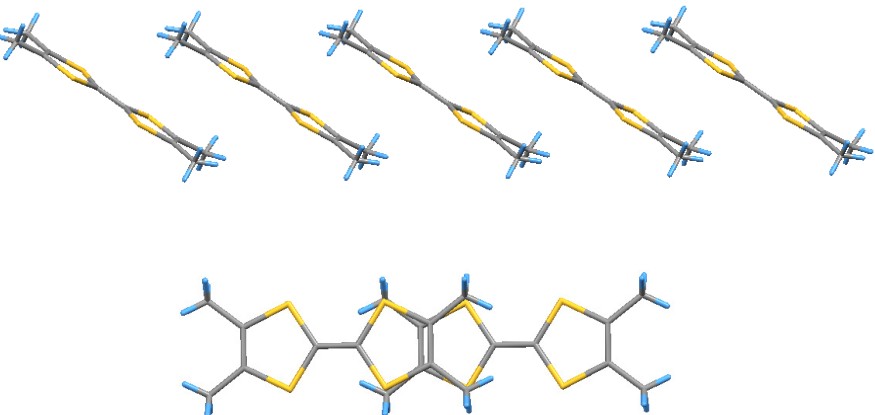

**Figure 5.** A stack and radical-cations overlapping in (**2**).

The Fe-C and Fe-B bond lengths are 2.0790(9)–2.1001(8) and 2.1001(8)–2.1494(8) Å, correspondingly, and the overlapping values are due to the statistical disordering of carbon and boron atoms in the dicarbollide ligands. The distances from the iron atom to the $C_2B_3$ faces of the dicarbollide ligands are equal to 1.53 Å, which is close to the distances in analogous salts of the iron bis(dicarbollide) anion [19,24,25]. The dicarbollide ligands are turned relative to each by 180°, forming the *transoid* conformation. The $C_2B_3$ faces are parallel by symmetry conditions.

According to the electric conductivity measurements, compound 2 is an insulator with conductivity ~$10^{-10}$ Ohm$^{-1}$cm$^{-1}$. The low value of electroconductivity is in an agreement with the 1:1 stoichiometry and non-layered structure of the salt, as well as with the inclination angle of the radical-cations in the stack, at which there is only slight overlap between neighboring radical-cations.

The $(TMTSF)_2[3,3'\text{-}Cr(1,2\text{-}C_2B_9H_{10})_2]$ (3) crystals are isostructural to $(TMTSF)_2[3,3'\text{-}Co(1,2\text{-}C_2B_9H_{11})_2]$ and $(TMTSF)_2[3,3'\text{-}Fe(1,2\text{-}C_2B_9H_{11})_2]$ salts studied earlier, containing cobalt and iron bis(dicarbollide) anions [18,19]. The crystal structure of 3 is formed by the TMTSF cation in a general position and the $[3,3'\text{-}Cr(1,2\text{-}C_2B_9H_{11})_2]^-$ anion in a special centrosymmetrical position (Figure 6). Compound 3 possesses a structure (Figures 7 and 8) where the TMTSF$^{+\bullet}$ radical-cations and anions form staggered stacks. The distances between the averaged planes of the TMTSF donors in the dimers are 3.70 and 3.73 Å, and the dihedral angle between the planes is 0° by symmetry conditions.

The Cr-C and Cr-B bond lengths are 2.175(7)–2.176(7) and 2.226(8)–2.277(8) Å, correspondingly. The distances from the chromium atom to the $C_2B_3$ faces of the dicarbollide ligands are equal to 1.68 Å, and the dicarbollide ligands in the $[3,3'\text{-}Cr(1,2\text{-}C_2B_9H_{10})_2]^-$ anion are turned relative to each other by 180°, forming the *transoid* conformation. The $C_2B_3$ faces are parallel to each other by the symmetry conditions.

The electroconductivity measurements have revealed that compound 3 in the range of 41–195 K behaves like a dielectric. However, above 195 K, the delocalization of the positive charge disappears due to the numerous intermolecular S . . . S contacts and an inconspicuous dielectric–semiconductor structural phase transition occurs, caused by charge ordering: stacks contain both TMTSF molecules and TMTSF radical-cations. The room temperature electric conductivity $\sigma_{293} = 5{\cdot}10^{-3}$ Ohm$^{-1}$cm$^{-1}$ and activation energy $E_a \cong 0.04$ eV (Figure 9). It should be noted that analogous salts $(TMTSF)_2[3,3'\text{-}Co(1,2\text{-}C_2B_9H_{11})_2]$ and $(TMTSF)_2[3,3'\text{-}Fe(1,2\text{-}C_2B_9H_{11})_2]$ were characterized by electroconductivity values $\sigma_{293}$ of 15 and 0.1 Ohm$^{-1}$cm$^{-1}$, correspondingly [18,19].

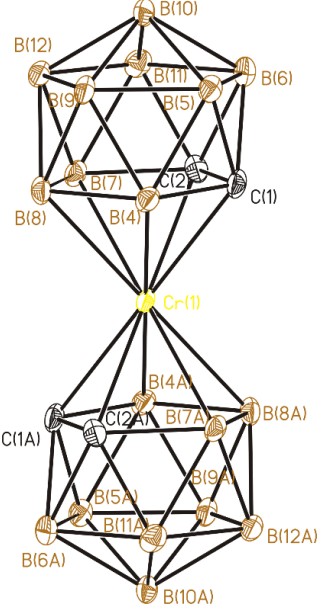

**Figure 6.** Anion in (**3**). Thermal ellipsoids are given at 30% probability level.

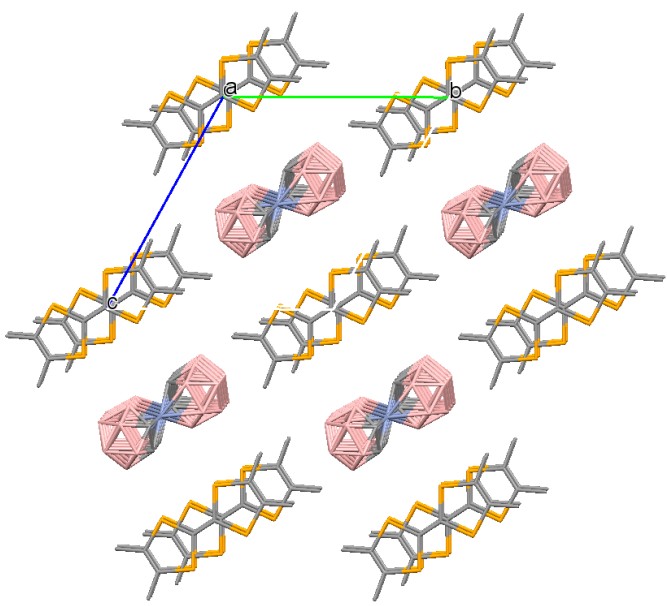

**Figure 7.** Crystal packing fragment of (**3**). A view along the *a* axis. The unit cell is outlined. H atoms are omitted for clarity.

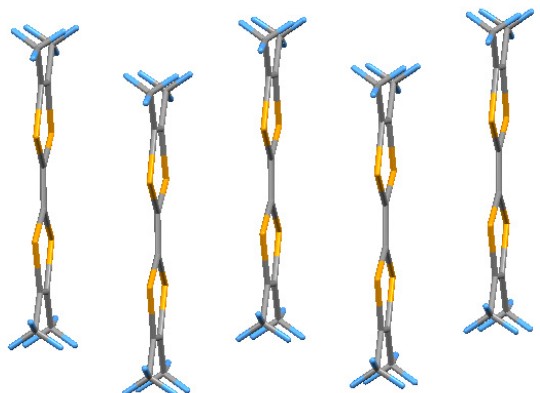

**Figure 8.** A stack of radical-cations in (**3**).

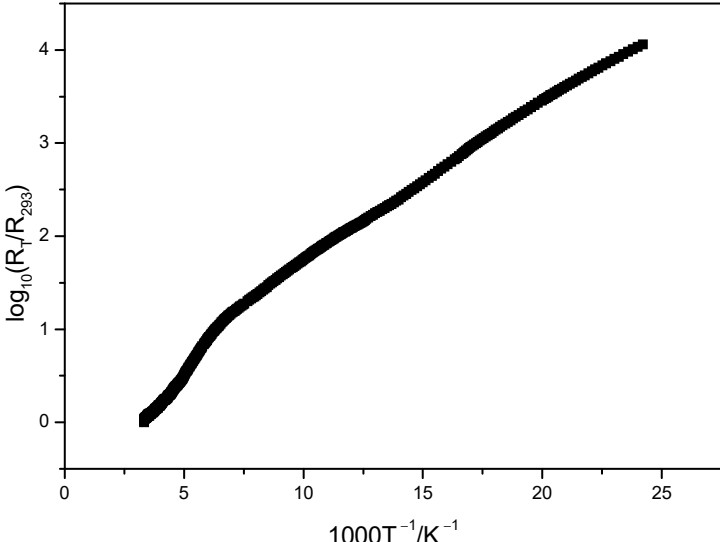

**Figure 9.** Temperature dependence of electrical resistivity of (**3**).

In conclusion, new salts with the TMTTF and TMTSF radical-cations and metallacarborane anions (TMTTF)[3,3′-Cr(1,2-$C_2B_9H_{11}$)$_2$] (**1**), (TMTTF)[3,3′-Fe(1,2-$C_2B_9H_{11}$)$_2$] (**2**), and (TMTSF)$_2$[3,3′-Cr(1,2-$C_2B_9H_{11}$)$_2$] (**3**) were electrochemically synthesized and investigated. Their crystal structures were determined by X-ray study and electroconductivities were measured. Salts (**1**) and (**2**) are insulators, which is explained by the 1:1 stoichiometry and the absence of an extended network of interdonor interactions, whereas (**3**) is a semiconductor at room temperature with electroconductivity $\sigma_{293}$ = 5·$10^{-3}$ $Ohm^{-1}cm^{-1}$, which is lower than in (TMTSF)$_2$[3,3′-Fe(1,2-$C_2B_9H_{11}$)$_2$] and (TMTSF)$_2$[3,3′-Co(1,2-$C_2B_9H_{11}$)$_2$] salts (electroconductivity values $\sigma_{293}$ of 0.1 and 15 $Ohm^{-1}cm^{-1}$, correspondingly). The tendency of a rise in conductivity (5·$10^{-3}$ < 0.1 < 15) is apparently connected with decreasing the cation size in the order $Cr^{3+}$ > $Fe^{3+}$ > $Co^{3+}$ [26], which leads to decreasing the corresponding metallacarborane anion size and, in turn, to unit cell compression and a tighter radical-cation packing of the salts.

**Supplementary Materials:** Details of experimental data including synthesis of the title compounds, their X-ray diffraction studies, and electric resistivity measurements are available online at https://www.mdpi.com/article/10.3390/cryst11091118/s1.

**Author Contributions:** Synthesis, D.M.C., I.B.S., I.D.K., G.G.A., E.V.S.; measurements, L.I.B., D.N.P., T.N.S.; X-ray diffraction study, G.V.S.; data analysis and writing, O.N.K.; research conception, V.I.B., O.A.D. All authors have read and agreed to the published version of the manuscript.

**Funding:** This research received no external funding.

**Institutional Review Board Statement:** Not applicable.

**Informed Consent Statement:** Not applicable.

**Data Availability Statement:** The crystallographic data for this paper (the CCDC numbers 2091714, 2091713, 2091715 for (**1**), (**2**), (**3**), respectively) can be obtained free of charge via www.ccdc.cam.ac.uk/data_request/cif (accessed on 11 August 2021).

**Acknowledgments:** This work was partly performed in accordance with the state task of the Institute of Problems of Chemical Physics, Russian Academy of Sciences, State Registration No. AAAA-A19-119092390076-7. Synthesis and characterization of the starting metallacarboranes were performed at A.N. Nesmeyanov Institute of Organoelement Compounds, Russian Academy of Sciences with support of the Ministry of Science and Higher Education of the Russian Federation.

**Conflicts of Interest:** The authors declare no conflict of interest. The funders had no role in the design of the study; in the collection, analyses, or interpretation of data; in the writing of the manuscript, or in the decision to publish the results.

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
