# Peer review of "New Radical-Cation Salts Based on the TMTTF and TMTSF Donors with Iron and Chromium Bis(Dicarbollide) Complexes: Synthesis, Structure, Properties [†]"

_crystals, doi:10.3390/cryst11091118_

Round 1

Reviewer 1 Report

This contribution by D. M. Chudak et al. reports 3 new radical cation salts based on TMTTF and TMTSF donors with Iron and Chromium Bis(Dicarbollide) Complexes as anions.

These compounds are the result from the work of the team during the past years  in exploring salts of TTF derivatives with metallocarborane anions, a type of anions that has been lees explored, therefore making interesting this type of salts and justifying the publication their preparation and characterization . Two salts have a 1:1 stoichiometry and are insulating and the last one with 2:1 stoichiometry, is a poor semiconductor.

While the results are generally well described in an easy to follow manner, the manuscript would benefit from a throughout revision of its English style before publication. A few specific points are the following.

  • in line 70 page 2 where it reads a “bath” conformation, this should be replaced by “boat” conformation.
  • in line 88 page 2 where it reads “… are known, studied us before” needs to be rephrased probably with “ ... studied by us … “
  • in line 118 page 5 it is quite confusing where it reads “ … low overlapping occurs between the neighboring radical-cations and between the S atoms orbitals”. Besides the poor English of this phrased there is a confusion of two connected concepts; The overlap between S atoms orbitals, is obviously part of the intermolecular molecular overlap between cations.
  • In page 6 of the text it is mentioned that the electrical conductivity presents a semiconducting behavior with an activation energy Ea ~ 0.04 eV in the range of 41 - 293 К . However this is an oversimplification since Figure 9 shows that the activation energy Ea is not at all constant in this temperature rage and clearly presents a maximum at ~200-190K, possibly associated with some structural change. Authors cannot ignore this and should comment on this.
  • in line 154 page 7, where it is claimed that the low conductivity “…  is connected with 1:1 stoichiometry and the absence of classical conducting layers or stacks in the structure. Here the “absence of classical conducting layers or stacks in the structure” should be better replaced by an “absence of an extended network of interdonor interactions”.

Author Response

Thank you very much for your attention to our work. We have tried to take into account all your comments and advices. The text of the manuscript has been carefully proofread and the English has been verified to the best of our ability. Special thanks for your comment regarding Fig. 9 (Page 6).

 All changes in the text of the manuscript are highlighted in yellow.

Reviewer 2 Report

The authors report here on new radical cation salts of TMTTF and TMTSF with Cr(III) and Fe(III) metallacarborane monoanions. Although the physical properties are not particularly interesting, the crystal structures were properly determined and discussed. The work appears somewhat incremental with respect to previous reports of the authors, and the reader cannot really disclose the interest of the new salts. With some revisions the manuscript could become worthy of publication in Crystals.

Some revisions to be addressed:

1) The Introduction is very short and does not highlight properly the motivation and interest of the work. Why the use of metallacarboranes in TTF based radical cation salts is interesting? The authors took great care to cite their own previous work but not at all important contributions in the field. I don’t agree, for example, with the statement “very little attention was paid to radical-cation salts based on TMTTF [10,11] and TMTSF [12,13]”. Maybe the authors wanted to stress out that not many salts of TMTTF and TMTSF contain metallacarborane anions. This should be clarified. However, I suggest the authors to put their study in a broader perspective and to add, for example, the following references on TMTTF and TMTSF radical cation salts: 1) Crystals 2021, 11, 386; 2) Crystals 2021, 11, 838; 3) Crystals 2020, 10, 1119; 4) Cryst. Growth Des. 2020, 20, 6777−6786; 5) Chem. Sci. 2020, 11, 10078–10091; 6) New J. Chem. 2020, 44, 15538–15548, and so on…

2) page 1 in the Introduction, I would not say the donors cited by the authors are “exotic”.

3) page 2, lines 86-88, the last sentence is unclear.

4) the authors don’t say anything about the magnetic properties of the salts.

5) what was the temperature in the electrocrystallization experiments?

6) I do not see why the structural figures are not coloured.

7) English should be polished.

Author Response

Thank you very much for your attention to our work. We have tried to take into account all your comments and wishes. In particular, the introduction has been expanded and recommended literature references have been added. The text of the manuscript has been carefully proofread and the English has been verified to the best of our ability. The electrocrystallization temperature was indicated in the Experimental part. All changes in the text of the manuscript are highlighted in yellow.

In response to the question about the absence of magnetic measurements, we must admit this disadvantage, which is a weak point. This is due to the lack of the necessary equipment at our disposal, as well as the unexpected death of one of the members of our international team, who was organizing magnetic measurements, due to the COVID pandemic. Unfortunately, we have not yet been able to restore the broken scientific contacts, but we hope that we will be able to do this in the near future, since we ourselves are very interested in this.

As for the colored figures of the structures, we have not used them before because of the relatively small number of different atoms and the simplicity of the cationic and anionic components. However, this is an interesting and worthwhile idea and we hope to use it in the future.

Round 2

Reviewer 2 Report

The authors did their best to address the reviewer's requests. While I can admit that magnetic measurements can be an issue depending on the available equipment and qualified collaborators, I still do not understand why the authors do not produce more enjoyable colour figures instead of the black and white ones. No additional publication fees are requested for the use of colours and the clarity of the figures is much improved although there are not many different atoms in the molecules.

Author Response

Thank you for your interest in our work. We made colored structure figures according to your recommendations.